# Neuroendocrine-Related Circulating Transcripts in Small-Cell Lung Cancers: Detection Methods and Future Perspectives

**DOI:** 10.3390/cancers13061339

**Published:** 2021-03-16

**Authors:** Lucia Anna Muscarella, Tommaso Mazza, Federico Pio Fabrizio, Angelo Sparaneo, Vito D’Alessandro, Antonio Tancredi, Domenico Trombetta, Flavia Centra, Silvana Pia Muscarella, Concetta Martina Di Micco, Antonio Rossi

**Affiliations:** 1Laboratory of Oncology, Fondazione IRCCS Casa Sollievo della Sofferenza, 71013 Foggia, Italy; fp.fabrizio@operapadrepio.it (F.P.F.); a.sparaneo@operapadrepio.it (A.S.); d.trombetta@operapadrepio.it (D.T.); f.centra@operapadrepio.it (F.C.); 2Unit of Bioinformatic, Fondazione IRCCS Casa Sollievo della Sofferenza, 71013 Foggia, Italy; t.mazza@operapadrepio.it; 3Division of Internal Medicine, Fondazione IRCCS Casa Sollievo della Sofferenza, 71013 Foggia, Italy; v.dalessandro@operapadrepio.it; 4Unit of Thoracic Surgery, Fondazione IRCCS Casa Sollievo della Sofferenza, 71013 Foggia, Italy; a.tancredi@operapadrepio.it; 5Department of Diagnostic and Interventional Radiology, Azienda Ospedaliera per l’Emergenza Cannizzaro, 95126 Catania, Italy; silvana.muscarella@aoec.it; 6Unit of Oncology, Fondazione IRCCS Casa Sollievo della Sofferenza, 71013 Foggia, Italy; doctor.dimicco@gmail.com (C.M.D.M.); arossi_it@yaoo.it (A.R.)

**Keywords:** SCLC subtypes, CTC, neuroendocrine transcripts, nELAVs, SSTRs, SCG3, DLL3, SYP, CHGA, proGRP

## Abstract

**Simple Summary:**

The recent implementation of techniques to study circulating tumor cells allowed a rapid increase in knowledge about the molecular basis of Small-Cell Lung Cancer (SCLC), which appears to be more heterogeneous and dynamic than expected. Here, we present a summary of current knowledge and new findings about some of the neuroendocrine-related transcripts expressed in SCLC patients that could offer a great opportunity in distinguishing and managing different SCLC phenotypes.

**Abstract:**

No well-established prognostic or predictive molecular markers of small-cell lung cancer (SCLC) are currently available; therefore, all patients receive standard treatment. Adequate quantities and quality of tissue samples are frequently unavailable to perform a molecular analysis of SCLC, which appears more heterogeneous and dynamic than expected. The implementation of techniques to study circulating tumor cells could offer a suitable alternative to expand the knowledge of the molecular basis of a tumor. In this context, the advantage of SCLC circulating cells to express some specific markers to be explored in blood as circulating transcripts could offer a great opportunity in distinguishing and managing different SCLC phenotypes. Here, we present a summary of published data and new findings about the detection methods and potential application of a group of neuroendocrine related transcripts in the peripheral blood of SCLC patients. In the era of new treatments, easy and rapid detection of informative biomarkers in blood warrants further investigation, since it represents an important option to obtain essential information for disease monitoring and/or better treatment choices.

## 1. Introduction

Small-cell lung cancer (SCLC) is an aggressive type of cancer with an incidence of about 13–15% among lung cancers and has a very poor prognosis due to its rapid development of resistance to chemo- and radiotherapies [1]. Unlike the increase in personalized approaches to the clinical care of patients with non-small-cell lung cancer (NSCLC), clinical protocols for SCLC still mainly depend on the stage of the disease, prior therapies, and lack of specific molecular support. This approach was mainly due to the idea of SCLC as a monolithic entity with common genetic features [2], which was strictly linked to the lack of an adequate quantity of tissue samples in this inoperable class of patients, for the lack of a clear and comprehensive biological profile presented an obstacle [3,4].

Due to technological advances, many large studies were published, supporting a new and more complex definition of SCLC. The studies suggested the existence of four biologically distinct subtypes of SCLC associated to specific therapeutic vulnerabilities and outcomes. Each of these subtypes is defined by its inter tumor expression levels of the four key transcription regulators: Achaete–Scute Family BHLH Transcription Factor 1 (ASCL1), Neuronal Differentiation 1 (NEUROD1), POU Class 2 Homeobox 3 (POU2F3), and yes–associated protein 1 (YAP1) [5]. The results from studies on genetically engineered mouse models of SCLC suggested that different neuroendocrine and non-neuroendocrine tumor cells could coexist in the same tumor mass, guiding its ability to rapidly evolve under selective pressure induced by a specific treatment [5]. Given the availability of highly sensitive and high-throughput molecular technologies, the peculiar blood and lymphatic spreading of SCLC and some of their neuroendocrine features became critical features that offered the opportunity to noninvasively access SCLC molecular markers. Tumor cells that circulate in the blood (circulating tumor cells (CTCs)) have a widely reported prognostic value in many tumors. They offer the advantage of extracting nucleic acids from the nucleated cell fraction of peripheral blood to obtain information, such as the expression levels of specific molecular SCLC markers [6,7]. This could help to obtain both a more molecular background that could contribute to differentiating early or metastatic SCLC from NSCLC and to easily define the specific subtypes of SCLC, thus improving disease management [5,8].

The noninvasive strategy to monitor disease was purposed for SCLC patients and was successfully assessed in some papers and was confirmed on a few patient cohorts [9]. In this review, we summarize the scientific knowledge about a group of ectopic neuroendocrine tumor-associated transcripts of SCLC. These transcripts were detected in the whole peripheral blood (PB) of SCLC patients by highly sensitive techniques, and were suggested as surrogates and noninvasive biomarkers of CTCs. Specifically, this review covers all published data in the field about somatostatin receptors (SSTRs), neuronal embryonic lethal, abnormal vison, Drosophila-like proteins (nELAVs), synaptophysin (SYP), chromogranin A (CHGA), delta-like ligand 3 (DLL3), pro-bombesin-like peptide (ProGRP), and secretogranin III (SGC3). Some of these transcripts are reported to be strictly related to one of the four proposed SCLC subtypes. By consequence, their detection in blood could represent an option to rapidly profile SCLC patients. To confirm the published data about SCG3, we described the optimization of a simple and real time quantitative PCR approach as a noninvasive methodology to detect this transcript in the PB of SCLC patients.

## 2. Circulating SSTRs Transcripts

SSTRs are G-protein-coupled receptors encoded by five different SSTR1-5 genes, whose expression on the cell surface has been demonstrated both in SCLC cell lines [10,11] and tissues by using imaging studies and positron emission tomography (PET)/CT [12,13,14]. Somatostatin regulates the complex machinery of hormone secretion, and its activity depends on specific SSTRs receptors located on the surface of cells. The SSTRs expression on SCLC cells provided the molecular basis for the successful use of radiolabel synthetic analogs, such as the ^68^Ga/^177^Lu/^90^Y-labeled compounds DOTATATE/-TOC and ^111^In--DTPA-D-Phe1-octreotide scan, which binds with high affinity to SSTR2a, and with moderate affinity to SSTR3 and SSTR5, to detect tumors [15,16,17,18]. A recent study described a strong linkage between SSTR2 and NEUROD1 expressions, suggesting that these two markers could coexist in the same populations of the SCLC subtype within the same tumor [5,19]. 

Several studies applied autoradiography and molecular biology techniques, such as real-time polymerase chain reaction (RT-PCR), in situ hybridization (ISH), Northern blotting, and immunohistochemistry (IHC), to determine SSTRs expression in endocrine tumors, with mixed success [13,20,21]. To overcome the main limitations of these techniques related to the heterogeneous SSTR distribution in tumors and the existing difference of expressions between the transcript (mRNA) and SSTR protein levels, a sensitive RT-qPCR approach was developed to measure the SSTRs transcripts levels in the PB of lung neuroendocrine tumor patients [22]. The assay showed a sensitivity of 86% for both SSTR2a and SSTR5, 83% for SSTR2a, and 79% for SSTR5 in distinguishing the normal and pathological blood samples of SCLC patients. Compared with results from ^111^In-DTPA-D-Phe^1^-octreotide (OctreoScan), an overall 76% agreement was demonstrated. This correspondence is higher than those reported by Righi et al. obtained by comparing the OctreoScan results with IHC analysis of SSTRs proteins in neuroendocrine lung tumors [21]. 

No correlation with the disease outcome was reported, even though published studies confirmed that the standardized uptake value (SUV) measured with a PET/CT or histological SSTR2a expression had prognostic value [13]. Despite the gap in this information, the used RT-PCR technique has the main advantage of detecting a few tumor cells in one million normal peripheral blood cells with high sensitivity [23]. This could facilitate the monitoring of markers’ variations in terms of mRNA amount, also in biological fluid containing very low quantities of nucleic acids. The detection of the mRNA levels in circulating tumor cells by RT-qPCR was reported as a useful approach for the definition of minimal residual disease (MRD) in patients affected by neuroendocrine tumors [24]. However, studies on SCLC are lacking and could be useful to confirm the importance of SSTRs transcript detection in PB in this field.

Another consideration that empathizes the power of circulating SSTRs transcripts detection is the use of personalized peptide receptor radionuclide therapies to enhance its efficacy, also in combination treatments with other systemic therapies, such as radiosensitizing chemotherapy, DNA repair-modifying agents and immunotherapy [17,25]. The recent interest in SSTRs detection in SCLC was linked to the lutathera treatment: a beta-emitting, lutetium-177 labeled somatostatin approved for the treatment of neuroendocrine tumors [26]. Lutathera synergically acts with monoclonal antibodies interfering with the inhibitory-programmed death (PD)-1/PD-L1 pathway, approved for advanced SCLC previously treated with one or more platinum based chemotherapies. In the phase I study of CheckMate 032, it was reported that lutathera combined with nivolumab showed some antitumor activity, suggesting that the strong uptake of ^68^Gallium-DOTATATE may predict efficacy of combining lutathera with anti-PD-L1 therapy in SCLC patients [27]. Nevertheless, no definite conclusion regarding the efficacy of this approach can be drawn due to the small cohort numbers, thus further investigations are warranted.

## 3. Circulating nELAVs Transcripts

nELAV proteins, also named Hu antigens, are a family of antigens normally found in neurons. They are ectopically expressed in 90% of SCLC patients having paraneoplastic syndromes (PNSs), a group of rare disorders that can affect any part of the nervous system in patients with cancer [28,29]. PNSs frequently arise from an autoimmune response triggered by the ectopic expression of neuronal proteins in cancer cells. An onconeural immune response elicited by anti-Hu antibodies can also occur in SCLC patients without any evident neurological dysfunctions [30].

Antibodies against nELAV proteins could be detected in the serum of patients affected by SCLC using enzyme-linked immunosorbent assay (ELISA) [31,32,33]. Interestingly, a quantitative and high-sensitivity RT-qPCR was suggested as an alternative approach to detect SCLC cells in the PB of early-stage patients by measuring nELAV transcripts levels. Low levels of nELAVs expression were detected in the group of healthy blood donors for all three genes, so the authors established a cut-off value to distinguish normal and pathological samples. Of the samples, 24%, 52%, and 16% of samples scored positive for HuD, HuB, and HuC, respectively, whereas 20% of samples showed concordant results for both HuD and HuB markers. Overall, 56% of the analyzed tumors showed HuD or HuB transcripts above the cut-off values. The quantification of HuB, HuC, and HuD expressions in the PB of SCLC patients, by targeting the 3′ untraslated (UTR) regions, could be of interest to discriminate transcripts of the same gene family having similar sequences. However, this approach was reported in a single paper and additional studies with larger patient groups are required to confirm any experimental observations [34]. With the inclusion of immune-checkpoint inhibitors (ICIs) in the cancer treatment algorithm, the interest in PNSs and nELAV has increased since their link to immunity was demonstrated in SCLC patients. In several SCLC with anti-Hu antibodies, SCLC’s spontaneous regression without treatment has been reported, thus suggesting a host immune response directed against both cancer and the nervous system [35,36,37]. ICIs are associated with a considerably increased incidence in immunological toxicities than traditional anticancer therapies, including neurological immune-related adverse effects, which can manifest as PNSs [28]. Consequently, immunotherapy might increase the risk of PNSs in SCLC patients, since this type of cancer is frequently associated with these disorders. In this context, to determine whether the syndromes fulfil the criteria for a real PNS through the anti-Hu expression levels, noninvasive monitoring could be crucial, since each scenario has different prognostic and treatment implications for patients treated with ICIs [38,39].

## 4. Circulating SYP, CHGA, and DLL3 Transcripts

Synaptophysin (SYP) and chromogranin A (CHGA) are two well-known neuroendocrine markers currently used to confirm the histological diagnosis of SCLC by IHC [40]. The delta-like ligand 3 (DLL3) is an inhibitory ligand of Notch that has gradually gained interest since it became the therapeutic target of the antibody drug conjugate rovalpituzumab-tesirine (Rova-T) [41]. DLL3 expression is regulated by ASCL1, a well-known transcription factor required for the proper development of pulmonary neuroendocrine cells of SCLC. DLL3 is one of the oncogenic drivers in SCLC, associated with neuroendocrine tumorigenesis, SCLC migration, and invasion through a mechanism that involves control of the epithelial–mesenchymal transition [42]. DLL3 expression showed temporal variation under platinum treatment, suggesting it could represent a suitable specific SCLC marker of the adaptive response to platinum-based agents [43,44,45]. The peculiar expression of DLL3 makes it suitable as a noninvasive marker of SCLC. DLL3 is highly expressed on the surface of SCLC cells and other neuroendocrine tumors, but it is minimally expressed in normal tissues [46,47]. A recent study published by Obermayr et al. [48] demonstrated that the DLL3 transcript can be detected together with SYP and CHGA in the CTC-enriched fraction of blood samples of SCLC patients using a combination of Parsortix’s technology selection and RT-qPCR (8% and 25%, respectively). DLL3 transcripts levels are a candidate prognostic marker, since DLL3-positive patients had a significantly shorter overall survival (OS) than those who were negative and the risk of dying earlier was higher in the first-cited group than the last. A positive detection of SYP and CHGA transcripts levels measured at different times was associated with a high risk of early death, irrespective of whether the samples were taken at primary diagnosis or at disease progression. Unfortunately, it was not possible to draw the same conclusions for the DLL3 marker because of the small number of patients [48]. In the same paper, the author used the approach to combine results from multiple neuroendocrine genes expression profiles to obtain a unique prognostic expression marker for the SCLC patients. Despite the utility of this approach in the PB, the technical procedure suffers from low sensitivity due to the need to split each sample into aliquots to analyze the expression of multiple genes individually. Considering the limitations of the work, the results reported by Obermayr et al. [48] are interesting since they highlight two important translational applications of DLL3, SYP, and CHGA detection in PB by RT-PCR. Firstly, the presence of these specific neuroendocrine transcripts, at any stage of disease, was associated with a worse outcome and could be proposed as a noninvasive method to investigate and support the identification of the ASCL1-SCLC subgroup of patients and monitor tumor resistance acquisition during treatments [49]. Secondly, patients’ stratification for personalized treatment options, such as Rova-T, were based on the analysis of targets, such as DLL3 in tissue samples, which were taken long before the disease progression occurred. In contrast, liquid biopsy samples could be taken right before the start of treatment, so that the detection of DLL3 mRNA in the PB of patients could provide a snapshot and dynamic analysis toward personalized treatments in SCLC patients.

## 5. Circulating ProGRP Transcript

ProGRP is the precursor protein of the bombesin-like peptide (GRP), a neuropeptide and autocrine growth factor expressed in nerve fibers, gastric tissue, brain, fetal lung, pulmonary carcinoid, and SCLC cells [50,51]. A recent study indicated that ProGRP, together with synaptophysin, chromogranin A, and ASCL1, were significantly decreased in a specific subset of TTF-1 negative SCLC samples. It was observed that an elevated expression of ASCL1 induced production of serum ProGRP, thus favoring the neuroendocrine differentiation of tumor cells [52].

Due to the lack of its expression in the hematopoietic or endothelial cells in blood vessels and in the epithelial cells of the human skin, proGCR was one of the first mRNA successfully detected in the blood of SCLC patients. Its expression was found in SCLC cell lines, blood and marrow samples, and pleural effusion at any stage SCLC using nested reverse transcriptase RT-PCR ProGRP. In contrast, the expression of ProGRP at the mRNA level was absent in the same body fluids of NSCLC and affected patients, patients with nonmalignant diseases, and healthy volunteers [53]. Recently, independent studies suggested the potential prognostic and predictive value of ProGRP mRNA blood levels of SCLC patients. ProGRP expression in the PB of SCLC patients significantly differed according to tumor size, disease stage, and distant metastasis, and correlated with the serum ProGRP protein levels. A strong, positive correlation between ProGRP mRNA and serum ProGRP protein levels was demonstrated in a large group of 122 naïve patients with SCLC [54]. Finally, a complementary role of ProGRP and neuron-specific enolase (NSE) serum level was reported as a useful support to diagnose, monitor the effects of chemotherapy, and predict survival of SCLC patients [54,55].

## 6. Circulating SCG3 Transcript: State of the Art

The RE-1 silencing transcription factor [56,57] is a repressor of neuronal and neuroendocrine genes in non-neuronal cells, which are differentially expressed in many solid tumors, including SCLC [58,59]. Promoter methylation [60] and alternative splicing processes contribute to producing a specific sNRSF/REST4 isoform lacking one repression domain [61,62,63,64,65,66], with a consequent increase in specific REST-associated transcripts [67]. REST recruits multiple groups of transcriptional cofactor complexes to coordinate progressive chromatin changes (EZH2), which ultimately switch off the expression of many target genes [68,69]. Among these, the expression profiling analysis in lung cancer cells experimentally depleted of REST revealed that SCG3 could represent indirect, noninvasive markers for patients affected by some lung tumors, such as SCLC [67]. High levels of SCG3 mRNA were successfully detected in the blood of SCLC patients and seemed to have a prognostic role in predicting worse survival and poor responses to chemotherapy [67]. However, this finding has not been replicated and the effective role of the circulating marker has not been validated in SCLC. In particular, it is unclear if SCG3 could be a marker of all SCLC cells or, more probably, a marker of a specific population of non-neuroendocrine SCLC cells. An increase in REST activity was shown to be associated with YAP1 expression, a marker strictly associated to a non-neuroendocrine variant of SCLC in an engineered mouse model of SCLC [70] and in CTC-derived xenografts (CDX) models [71]. The YAP1 non-neuroendocrine population of SCLC is also known to be more chemoresistant to common first-line chemotherapies, such as cisplatin [71]. As REST was first described as a transcriptional, dynamic repressor that blocks the neuroendocrine differentiation in non-neuronal cells of NSCLC, the most probable hypothesis could be that some transcripts restricted by REST, such as SCG3, should also represent specific markers of the YAP1-SCLC subpopulation cells and could serve as predictive markers of response to chemotherapy. 

## 7. SCG3 Transcript in PB of SCLC Patients: New Unpublished Findings

To corroborate the assumption that the SCG3 expression detected in the blood of SCLC patients is derived exclusively from tumor cells, we optimized a noninvasive methodology using real-time quantitative PCR to detect SCG3 transcript traces in the peripheral blood of SCLC patients.

### 7.1. Materials and Methods: Patients, Samples, RT-qPCR, and Statistical Methods

A total of 26 SCLC patients were enrolled in this study at the Italian hospital, Casa Sollievo della Sofferenza, for five years. For each patient, peripheral blood samples in PAX-gene Blood RNA Tubes (PreAnalytiX, Qiagen, Germantown, MD, USA) were collected at baseline. Clinic-pathological information, including sex, smoking habit, age at diagnosis, and disease stage, were collected for all patients [72] (Table 1). Blood samples were obtained from a similarly sized cohort (*n* = 34) of nonsmoking volunteers with no evidence of any clinically detectable disease at the time of the blood withdrawal. This control group was available from the Transfusional Centre at the same hospital and allowed the capturing of large effect sizes (Cohen’s d = 0.8, α error probability = 0.05). All the subjects provided written informed consent and the local ethical committee approved the study.

The total RNA from each PAX-gene Blood RNA Tubes was extracted using the PAX-gene Blood RNA kit (PreAnalytix), then quality and concentration were established using the 2100 Expert Analyzer (Agilent Technologies, Santa Clara, CA, USA). The total RNA (500 ng) was subjected to reverse transcription using the QuantiTect Reverse Transcription Kit (Qiagen, Germantown, MD, USA). SCG3 mRNA levels were assessed in triplicate using the previously published set of primers by Moss el al. [67] and SYBR-Green-based RT-PCR on 7900HT (Thermo Fisher, Applied Biosystems Division, Foster City, CA, USA). A relative quantification method with a standard curve was used and the transcript levels were normalized using the housekeeping gene glyceraldehyde phosphate dehydrogenase (GAPDH) [22]. The standard curves for RT-PCR were assessed by cloning PCR fragments for both SCG3 and GAPDH genes to obtain four plasmid serial dilutions (in the range of 1 × 10^6^ copies to ten copies). The mRNA levels were determined as the ratio of the target gene expression levels to the GAPDH expression. The patients’ baseline characteristics are reported as a median ± standard deviation or frequencies, and the percentages for continuous and categorical variables, respectively. The baseline comparisons were made using a chi-square test for the categorical variables and the Mann–Whitney U-test for the continuous variables. The discriminatory power of SCG3 levels was assessed by estimating the area under the receiver operating characteristic (ROC) curves, along with their 95% confidence intervals (CIs). A *p*-value < 0.05 was considered statistical significance. All analyses were performed using the R Foundation for Statistical Computing, version 3.6.

### 7.2. Expression of SCG3 Markers in Healthy Volunteers and SCLC-Affected Patients: Results and Discussion

The SCG3 expression levels in the PB samples from the 34 healthy blood donors (HBD) were matched with the SCLC patients for age, sex, and geographical location. Firstly, they were assessed and a median SCG3/GAPDH copy number ratio of 0.00 (IQR 0.00–5.21) was established. The optimal cut-off values for the SCG3 levels were introduced to distinguish between normal and pathological conditions using the ROC curve analyses and were equal to 5.91 (area under the curve (AUC) 0.64, 95% CI: 0.50–0.76) (Figure 1). To determine the SCG3 expression in the SCLC patients, the PB samples from 26 affected individuals were analyzed and the SCG3 levels were found to be statistically significantly higher in SCLC patients (median 1.32; interquartile range (IQR), 0.00–18.72) compared with samples from HBD (*p* = 0.038, Mann–Whitney *U*-test).

Based on the hypothesis that the SCG3 transcript levels may reflect the CTC burden in SCLC patients, we tested the disease outcome differences. No significant correlations were found between SCG3 expression in the PB of SCLC patients with their outcome.

Our results confirm the data previously reported by Moss et al. [67] about the ability to detect SGC3 transcript levels in the blood of SCLC patients and confirm the general good performance of RT-qPCR as a highly sensitive and noninvasive technique. Moreover, the correlation of the SCG3 transcript with SCLC condition corroborates the idea that it could represent a specific marker of tumor cells population. Unfortunately, we were not able to confirm that the SCG3 transcript levels in the PB were of prognostic significance in SCLC patients. We think that this point needs to be addressed in further studies with a larger number of SCLC patients, both at limited and extensive disease stages, since response to standard chemotherapy at present cannot be predicted at the time of diagnosis, but it is important in determining the survival of patients.

## 8. General Conclusions

One key insight emerging from the complementary human and mouse models studies is the classification of SCLC subtypes defined by a distinct gene expression profile that could impact treatment definition and planned clinical trials. The observed dynamic change in these new markers and the lack of available tissue during disease progression represent one of the main points to address for the SCLC patients care implementation. 

A large and compelling body of evidence has accumulated in the past decade and highlights the potential role of CTCs and circulating tumor nucleic acids (ctNAs) in the liquid biopsy to help in the clinical care of neoplastic patients.

In SCLC patients, the ability to detect SCLC biomarkers in blood, such as specific neuroendocrine-related transcripts, is poorly investigated, but could have multiple, potential applications in early detection, patient’s stratification, prognosis, or predicting the response to specific therapies (Table 2). Currently, the best application of CTC will be in preclinical studies to understand SCLC biology, chemoresistance, and, most importantly, the existing phenotypic subtypes in a noninvasive way. In this context, the features of SCLC rapidly disseminating in blood will be useful to set high-throughput methodologies to quickly profile SCLC patients by measuring some mRNAs levels, instead of measuring their protein levels, which could have a different turnover time. 

Few papers were published in this field, where data need to be confirmed and expanded. The detection of specific transcripts in blood related to distinct SCLC subtypes may help to define the vulnerabilities of patients and therapeutic targets that are the focus of recent, active, and planned clinical trials. All possible approaches that could help to clarify the differences in SCLC subtypes, including noninvasive detection of specific transcripts of tumor cells, may represent an important path forward in defining better treatments for SCLC. 

## Figures and Tables

**Figure 1 cancers-13-01339-f001:**
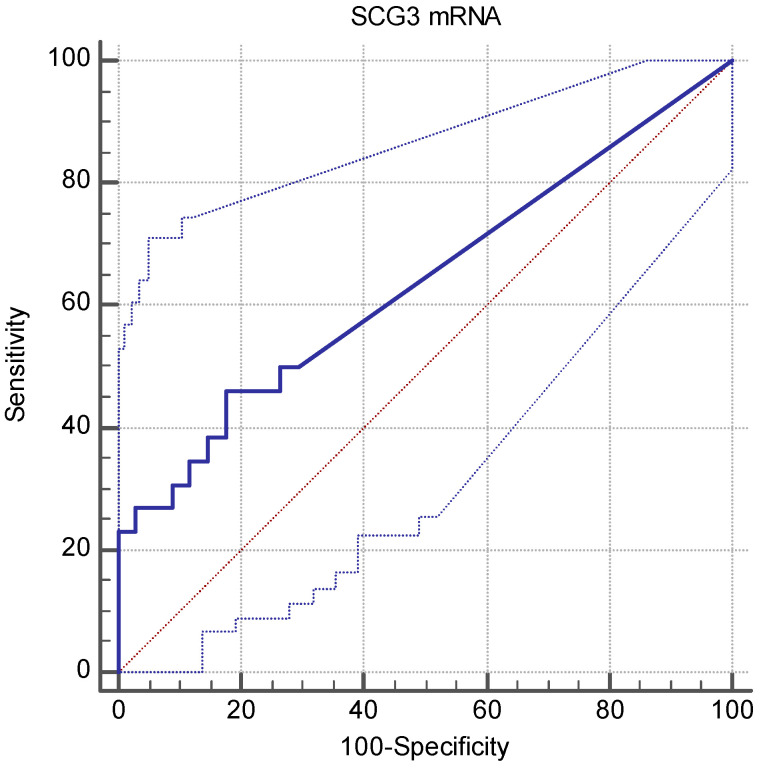
The area under the receiver operating characteristic (ROC) curve analysis of RT-qPCR. The ROC curve for the SCG3 mRNA was determined from the blood samples from healthy donors (*n* = 34) and cancer patients (*n* = 26). The area under the curve (AUC) is 0.64 (95% CI: 0.50–0.76).

**Table 1 cancers-13-01339-t001:** Clinical characteristics for the small-cell lung cancer (SCLC) patients.

Clinical Characteristics	Value
Number of Patients	26
Age (years)	
Median (range)	70 (42–82)
Sex, n (%)	
Male	26 (100)
Female	0 (0)
Smoking status, n (%)	
Former and current smoker	25 (96)
Never smoker	1 (4)
Pack/year (median ± SD)	50 ± 21
Disease stage, *n (%)*	
LD	7 (27)
ED	19 (73)
Median (range) survival (months)	7 (1–28)

LD, limited disease; ED, extensive disease.

**Table 2 cancers-13-01339-t002:** Circulating neuroendocrine-specific mRNAs detected in PB of SCLC patients.

Gene Symbol	Molecular Assay	Cohort Size (*N*)	% Positive Patients	Results	Refs
SSTR2a	RT-qPCR	14	93% (13/14)	The detection in PB was associated to SCLC	[22]
				The positivity in PB was correlated to OctreoScan captation	[22]
SSTR5	RT-qPCR	14	8/14 (57%)	The detection in PB was associated to SCLC	[22]
				The positivity in PB was correlated to OctreoScan captation	[22]
HuB (ELAVL2)	RT-qPCR	25	13/25 (52%)	The detection in PB was associated to SCLC	[34]
				The HuB/HuD positivity in PB was observed in LD SCLC stage	[34]
HuC (ELAVL3)	RT-qPCR	25	4/25 (16%)	The detection in PB was associated to SCLC	[34]
HuD (ELAVL4)	RT-qPCR	25	6/25 (24%)	The detection in PB was associated to SCLC	[34]
				The HuD/HuB positivity in PB was observed in LD SCLC stage	[34]
SYP	microfluidic Parsortix/RT-qPCR	51	11/51 (21.6%)	The detection in PB was associated to SCLC	[48]
				The SYP/CHGA positivity was associated to shorter OS	[48]
CHGA	microfluidic Parsortix/RT-qPCR	51	6/51 (12%)	The detection in PB was associated to SCLC	[48]
				The CHGA/SYP positivity associated to shorter OS	[48]
DLL3	microfluidic Parsortix/RT-qPCR	51	4/51 (7.8%)	The detection in PB was associated to SCLC	[48]
				The positivity in PB was associated to shorter OS	[48]
ProGRP	RT-qPCR	32	16/32 (50%)	The positivity in PB was associated to SCLC condition	[53]
	RT-qPCR	122	nr	The detection in PB was associated to SCLC	[54]
				The positivity in PB was correlated to ProGRP serum positivity	[54]
				The positivity in PB was correlated to tumor size, disease stage and distant metastasis	[54]
SCG3	RT-qPCR	67	24/67 (36%)	The detection in PB was associated to SCLC	[67]
				The positivity in PB was associated to poor survival and worse treatment response.	[67]
SCG3	RT-qPCR	26	12/26 (46%)	The detection in PB was associated to SCLC	*present work*

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
