# Peer review of "Neuroendocrine-Related Circulating Transcripts in Small-Cell Lung Cancers: Detection Methods and Future Perspectives"

_cancers, 2021, doi:10.3390/cancers13061339_

Round 1
Reviewer 1 Report
With these changes, the manuscript is now acceptable for publication.
Author Response
To the Guest Editors of Cancers
Dr. Salvatore Tafuto
Prof. Dr. Alessandro Ottaiano
Prof. Dr. Vincenzo Quagliariello
Special Issue “Neuroendocrine Neoplasms: Current Challenges and Advances in the Biological Aspects, Diagnostic and Therapeutic Management”
Dear Editors,
We appreciate very much the time spent in reviewing our manuscript entitled “Neuroendocrine-related circulating transcripts in small cell lung cancers: detection methods and future perspectives" submitted for publication in Cancers Journal as review.
We have read each of the reviewer’s comments carefully and have modified the manuscript accordingly. All modifications made in the uploaded revised version of the manuscripts are highlighted in yellow.
Best Regards,
Lucia Anna Muscarella

Reviewer 2 Report
Many thanks to the authors for taking some some of the considerations I raised in my initial review.
I feel that the manuscript requires significant editing to ensure that the main messages of this review are presented in a clear manner. I have highlighted the areas for concern in the attached PDF document.
This editing would be required prior to acceptance for publication.

Author Response
Dear Reviewer,
We appreciate very much the time spent in reviewing our manuscript entitled “Neuroendocrine-related circulating transcripts in small cell lung cancers: detection methods and future perspectives" submitted for publication in Cancers Journal as review.
We have read each of the your comments carefully and have modified the manuscript accordingly. All modifications made in the uploaded revised version of the manuscripts are highlighted in yellow.
Best Regards

Round 2
Reviewer 2 Report
I thank the author's for carefully modifying the manuscript according to the suggestions I made.
This manuscript is now acceptable for publication.
This manuscript is a resubmission of an earlier submission. The following is a list of the peer review reports and author responses from that submission.
Round 1
Reviewer 1 Report
This manuscript is mainly a short review with a single experiment. The manuscript presents an overly simplistic view of SCLC presenting incorrectly that all SCLC are neuroendocrine. The English usage in the manuscript requires editorial attention. The English is very poor. Too many needless abbreviations are used. The manuscript resembles a term paper more than a manuscript.
Specific comments:
- In three public databases, the SCG3 gene expression is highest in lung cancers and in brain tumors. The highest expressing lung cancers are not limited to SCLC. The highest expressing lung cancer in the CCLE is a squamous cell lung cancer.
- The current literature describes 4 subtypes of SCLC, this needs to be added to the manuscript.
- The manuscript presents a negative finding which is very likely correct.
Reviewer 2 Report
The authors provided a contemporary review of the blood markers associated with SCLC. In this review, the authors were critical of the sensitivity of methods used for analysing these blood markers and discussed the various treatment strategies associated with each blood marker. The authors also presented some of their own data in the manuscript around the detection of SCG3 transcripts in peripheral blood of SCLC patients. This was previously done by Moss et al. (2009), however, the manuscript aims to replicate this finding. The authors successfully detected SCG3 mRNA transcripts in 12/26 (46%) of patients compared to 24/67 (36%) found in Moss et al. (2009). Unfortunately, the manuscript did not find any significant correlations between SCG3 expression to patient outcomes due to the low number of SCLC patients recruited (line 288). In comparison, Moss et al. (2009) were able to show prognostic value of SCG3 expression to patient responses. Overall, while I can appreciate studies that aim to replicate findings, this manuscript does not provide any new insights into SCG3 expression and patient outcomes not already highlighted in Moss et al. (2009). Thus, for this reason the review is not suitable for publication in IJMS. Some major and minor comments are outlined below.
Major comments:
- In the abstract (lines 26-29) the author’s state that SCLC “has the advantage to express neuroendocrine markers that make the SCLC phenotype distinguishable from other normal or neoplastic cells in the lung”. While this statement is largely true, there is increasing data to suggest that SCLC exhibits a high degree of heterogeneity with not all tumour cells expressing high levels of neuroendocrine makers (Rudin et al. Nature Reviews Cancer 2019). This heterogeneity needs to be considered when discussing the utility of neuroendocrine transcripts in the blood as clinical biomarkers.
- Remove “state-of-the-art” at the end of each title header.
- Section entitled: SCG3 transcript in PB of SCLC patients: state-f-the-art: Further details (e.g. cohort size) on the study discussed on lines 223-225 should be included, given that this forms the basis of the additional (new) data presented in this review.
- Could the authors address the gender bias in their study (26 males, 0 females). It is known that SCLC affects both men and women in equal proportions. The statistical significance of findings should also be discussed/considered based on the small patient cohort size of this study (see: Dela Cruz, C.S., Tanoue, L.T., and Matthay, R.A. (2011). Lung Cancer: Epidemiology, Etiology, and Prevention. Clin Chest Med 32).
- The use of immune-checkpoint inhibitors (ICIs) in SCLC treatment is still contentious due to the mixed results surrounding PD-L1 expression (Yasuda et al., 2018). Could the authors comment on whether ICIs highlighted in the seminal Phase III CASPIAN study would benefit SCLC patients that develop paraneoplastic syndromes?
Paz-Ares, L., Dvorkin, M., Chen, Y., Reinmuth, N., Hotta, K., Trukhin, D., Statsenko, G., Hochmair, M.J., Özgüroğlu, M., Ji, J.H., et al. (2019). Durvalumab plus platinum–etoposide versus platinum–etoposide in first-line treatment of extensive-stage small-cell lung cancer (CASPIAN): a randomised, controlled, open-label, phase 3 trial. The Lancet 394, 1929–1939.
Yasuda, Y., Ozasa, H., and Kim, Y.H. (2018). PD-L1 Expression in Small Cell Lung Cancer. Journal of Thoracic Oncology 13, e40–e41.
- It is known that circulating tumour cells have acquired metastatic potential and have divergent neuroendocrine and non-neuroendocrine populations (Stewart et al., 2020; Pearsall et al., 2020). Given the heterogenous nature of SCLC with multiple transcriptional subtypes, could the authors clarify why a “pre-selection of CTCs that express neuroendocrine markers could be useful in this context to increase the sensitivity and specificity of neuroendocrine SCLC-specific transcript levels” (Line 304-305) is necessary?
Pearsall, S.M., Humphrey, S., Revill, M., Morgan, D., Frese, K.K., Galvin, M., Kerr, A., Carter, M., Priest, L., Blackhall, F., et al. (2020). The rare YAP1 subtype of Small Cell Lung Cancer revisited in a biobank of 39 Circulating Tumour Cell Patient Derived eXplant models (CDX): A brief report. Journal of Thoracic Oncology 0.
Stewart, C.A., Gay, C.M., Xi, Y., Sivajothi, S., Sivakamasundari, V., Fujimoto, J., Bolisetty, M., Hartsfield, P.M., Balasubramaniyan, V., Chalishazar, M.D., et al. (2020). Single-cell analyses reveal increased intratumoral heterogeneity after the onset of therapy resistance in small-cell lung cancer. Nature Cancer 1–14.
- Increase in REST activity has been shown to be associated with YAP1 expression, a non-neuroendocrine variant of SCLC, in a genetically engineered mouse model of SCLC (Ireland et al., 2020) and in CTC-derived xenografts (CDX) models (Pearsall et al., 2020). Moreover, this YAP1 non-neuroendocrine population is also known to be more chemoresistant to common first-line chemotherapies such as cisplatin (Pearsall et al., 2020). Could the authors comment on how decrease in REST, with an increase in SCG3, reflect a worse survival and poor response to chemotherapy (Line 224)? Is SCG3 reflective of a neuroendocrine population, but not a non-neuroendocrine population? This needs to be clarified.
Ireland, A.S., Micinski, A.M., Kastner, D.W., Guo, B., Wait, S.J., Spainhower, K.B., Conley, C.C., Chen, O.S., Guthrie, M.R., Soltero, D., et al. (2020). MYC Drives Temporal Evolution of Small Cell Lung Cancer Subtypes by Reprogramming Neuroendocrine Fate. Cancer Cell.
- Section entitled: SCG3 transcript in PB of SCLC patients: state-f-the-art: Further details (e.g. cohort size) on the study discussed on lines 223-225 should be included, given that this forms the basis of the additional (new) data presented in this review.
- Given that expression of neuroendocrine markers are not observed to the same degree in all patients (Table 1), can the authors also discuss about the utility of a NE/non-NE expression panel, and whether this may then account for possible intertumoral heterogeneity across patient cohorts. Possibly, elevating the robustness of this biomarker approach.
- This manuscript would benefit from scientific editing for grammar.
Additional comments:
- The authors could consider soliciting the assistance of scientific writers to assist with the editing of this manuscript.
- Line 66: summarized should read “summarize”
- Lines 66-69: “In this concise review…of these patients”. Sentence is too long, consider breaking this up to enhance the message of this review.
- Lines 69, 103: just should be changed to “recently”.
- Lines 119-120: “Anyhow, no definite conclusion….need further investigations” this sentence should be altered to “No definite conclusion regarding the efficacy can be drawn, due to the small cohort numbers, thus further investigations are warranted.”
- Line 131: Anyhow, should be replaced “Interestingly” is a suggested alternative.